# HDAC Inhibition Restores Response to HER2-Targeted Therapy in Breast Cancer via *PHLDA1* Induction

**DOI:** 10.3390/ijms24076228

**Published:** 2023-03-25

**Authors:** Natasha S. Clayton, Edward P. Carter, Abbie E. Fearon, James A. Heward, Lucía Rodríguez Fernández, Lina Boughetane, Edmund H. Wilkes, Pedro R. Cutillas, Richard P. Grose

**Affiliations:** 1Centre for Tumour Biology, Barts Cancer Institute, Queen Mary University of London, London EC1M 6BQ, UK; 2Centre for Haemato-Oncology, Barts Cancer Institute, Queen Mary University of London, London EC1M 6BQ, UK

**Keywords:** PHLDA1, TDAG51, HER2, lapatinib, breast cancer, drug resistance

## Abstract

The downregulation of Pleckstrin Homology-Like Domain family A member 1 (PHLDA1) expression mediates resistance to targeted therapies in receptor tyrosine kinase-driven cancers. The restoration and maintenance of PHLDA1 levels in cancer cells thus constitutes a potential strategy to circumvent resistance to inhibitors of receptor tyrosine kinases. Through a pharmacological approach, we identify the inhibition of MAPK signalling as a crucial step in *PHLDA1* downregulation. Further ChIP-qPCR analysis revealed that MEK1/2 inhibition produces significant epigenetic changes at the *PHLDA1* locus, specifically a decrease in the activatory marks H3Kme3 and H3K27ac. In line with this, we show that treatment with the clinically relevant class I histone deacetylase (HDAC) inhibitor 4SC-202 restores PHLDA1 expression in lapatinib-resistant human epidermal growth factor receptor-2 (HER2)^+^ breast cancer cells. Critically, we show that when given in combination, 4SC-202 and lapatinib exert synergistic effects on 2D cell proliferation and colony formation capacity. We therefore propose that co-treatment with 4SC-202 may prolong the clinical efficacy of lapatinib in HER2^+^ breast cancer patients.

## 1. Introduction

Breast cancer has the second highest incidence of all cancer types worldwide and is the second most common cause of cancer death in women in the UK [1]. Approximately 15–20% of all breast cancer cases are characterised by the overexpression of human epidermal growth factor receptor-2 (HER2), which is associated with increased invasiveness and drug resistance [2]. A number of HER2-targeted therapies have been developed, including humanised monoclonal antibodies, tyrosine kinase inhibitors, and antibody–drug conjugates [3]. Although these anti-HER2 agents have been shown to significantly reduce disease recurrence when given in combination with chemotherapeutic agents, therapeutic resistance typically occurs within months of beginning treatment [4]. The reactivation of PI3K/AKT and MAPK signalling is thought to underpin resistance to HER2-targeted therapy [5]. However, strategies to circumvent this using combinations of PI3K- and MEK-targeted therapeutics have shown limited success due to severe dose-limiting toxicities [6]. Novel approaches are needed in order to safely circumvent resistance to anti-HER2 therapies. 

We previously identified that the downregulation of Pleckstrin Homology-Like Domain family A member 1 (PHLDA1) underpins resistance to receptor tyrosine kinase (RTK)-targeted therapy in both fibroblast growth factor receptor 2 (FGFR2)-mutant endometrial and HER2-amplified breast cancers [7]. Since PHLDA1 is known to negatively regulate AKT [7,8,9], we demonstrated that the depletion of PHLDA1 levels facilitates the resurgence in PI3K/AKT signalling observed previously in drug-resistant cells [5]. We have shown that the recovery of PHLDA1 levels, using a doxycycline-inducible expression construct, restores the sensitivity of drug-resistant breast cancer cell lines to targeted therapy [7]. Restoring PHLDA1 expression therefore represents an attractive prospect for circumventing drug resistance in RTK-driven cancers such as HER2^+^ breast cancer. 

A broad range of stimuli have been reported to increase *PHLDA1* transcription, including heat shock [10], UV radiation [11], and treatment with homocysteine [12]. Most relevant to our present investigation into the mechanisms of lapatinib resistance are data placing PHLDA1 transcription downstream of HER2-regulated signalling cascades, such as the PI3K/AKT, MAPK, and JAK/STAT pathways. The treatment of MCF-7 cells with heregulin has been shown to increase PHLDA1 expression via a mechanism dependent on ERK1/2 and AKT activity [8]. A role for MAPK signalling in PHLDA1 regulation is further supported by work showing a 21-fold increase in PHLDA1 expression in human mammary epithelial (HME16C) cells expressing the oncogenic RASV12 mutant, which was attenuated by treatment with a MEK1/2 inhibitor [13]. Other work using the macrophage cell line Raw264.7 showed the upregulation of PHLDA1 expression following treatment with the TLR2 agonist Pam3CSK4 via a mechanism dependent on the activity of JAK, STAT3, and ERK1/2 [14]. More recently, PHLDA1 expression was shown to be enhanced in MEK1 mutant cells [15]. 

In the present study, using an inhibitor-based approach, we identified a PKC/MAPK-mediated mechanism of PHLDA1 regulation in HER2^+^ breast cancer cell lines. We show that the inhibition of MAPK signalling leads to a decrease in activatory epigenetic marks at the *PHLDA1* locus and that PHLDA1 expression can be restored by treatment with the clinically relevant class I HDAC inhibitor, 4SC-202. Importantly, we demonstrate that when combined, 4SC-202 and lapatinib exert synergistic effects on HER2^+^ breast cancer cell growth. 

## 2. Results

### 2.1. PHLDA1 Expression Is Regulated Predominantly by PKC and MAPK Signalling

We have shown that treatment of HER2^+^ breast cancer cell lines with lapatinib leads to the downregulation of PHLDA1 [7]. Accordingly, matrix-embedded spheroids of the HER2^+^ breast cancer cell line HCC1954 exhibited growth arrest following treatment with lapatinib, which was diminished with PHLDA1 knockdown (Figure 1A–C and Appendix A). 

To dissect which specific signalling pathways downstream of HER2 regulate this response, HCC1954 HER2^+^ breast cancer cells were treated with a panel of inhibitors targeting the main downstream effectors of HER2: PI3K/AKT, MAPK, PLCγ, and JAK-STAT (Figure 1D). Treatment with the MEK1/2 inhibitor PD0325901 and the Raf inhibitor TAK632 led to a marked decrease in PHLDA1, supporting a role for the MAPK pathway in PHLDA1 downregulation (Figure 1E) [15]. PHLDA1 levels were also decreased following treatment with the JAK inhibitor ruxolitinib and with the allosteric AKT1/2 inhibitor AKT-VIII, yet the inhibition of PI3K activity using ZSTK474 did not elicit a similar response. 

The restoration of PHLDA1 expression in RTK-inhibitor-resistant cells using a doxycycline-inducible PHLDA1 expression construct has been shown to increase sensitivity to RTK-targeted therapy [7]. Compounds capable of augmenting PHLDA1 expression may therefore be of clinical importance. Since treatment with phorbol 12-myristate 13-acetate (PMA) stimulated the PHLDA1 expression in human T-cells [16,17], we investigated whether PMA treatment could reverse PHLDA1 downregulation in drug-resistant breast cancer cells. Two HER2^+^ breast cancer cell lines resistant to lapatinib, SKBR3^LapR^ and HCC1954^LapR^, were treated with PMA for 48 h. PMA increased the PHLDA1 levels in lapatinib-resistant cells via a mechanism dependent on the activity of PKC and ERK1/2 (Figure 2A), which were potently activated by PMA [18]. These findings were recapitulated in two FGFR2-driven endometrial cancer cell lines resistant to the FGFR inhibitor AZD4547 (Appendix A), signifying that this mechanism of PHLDA1 regulation in RTKi resistance is not limited to breast cancer. 

To determine whether ERK1/2 activity promotes *PHLDA1* transcription, HCC1954 cells expressing a constitutively active ERK2 construct, HA-ERK2 (GOF), were transfected with a *PHLDA1* luciferase reporter construct. The expression of HA-ERK2(GOF) led to a significant increase in luciferase activity relative to parental HCC1954 (Figure 2B). Equally, the protein levels of PHLDA1 were enhanced in cells expressing HA-ERK2 (GOF) (Figure 2C). 

### 2.2. PHLDA1 Expression Is Regulated Epigenetically

Whilst PMA served as a useful tool to highlight the role of PKC/MAPK signalling in the regulation of PHLDA1 levels, its well-documented tumour-promoting effects limited its use in our studies. To identify a therapeutically viable strategy to enhance PHLDA1 expression, we interrogated the mechanism of *PHLDA1* transcriptional repression. ChIP-PCR analysis revealed a significant decrease in active marks of transcription across the *PHLDA1* locus in HCC1954^LapR^ cells relative to parental controls (Figure 3A). Specifically, we observed a significant decrease in histone 3 lysine 4 tri-methylation (H3K4me3) and histone 3 lysine 27 acetylation (H3K27ac). A significant decrease in H3K4me3 and H3K27ac at the *PHLDA1* locus was also observed in MFE-296 endometrial cancer cells treated for seven days with the FGFR inhibitor PD173074 (Appendix A). To examine whether these epigenetic changes were mediated by the inhibition of MAPK signalling, HCC1954 cells were treated with the MEK1/2 inhibitor PD0325901 and subjected to ChIP-PCR analysis. Treatment with either 1 μM lapatinib or 200 nM PD0325901 for 72 h resulted in a significant decrease in both H3K27ac and H3K4me3 at the *PHLDA1* locus (Figure 3B).

To elucidate the mechanism through which MAPK signalling may lead to the observed epigenetic changes at the *PHLDA1* locus, we interrogated our phospho-proteomic timeline of acquired resistance in MFE-296 endometrial cells treated with the FGFR inhibitor PD173074 [7] (Appendix A). Of the 6706 phosphopeptides identified, 525 were significantly up- or downregulated in samples treated with PD173074 compared to the DMSO controls for at least one time point. Three epigenetic modifier proteins were differentially phosphorylated after seven days of PD173074 treatment (Appendix A). These include a reduction in the inhibitory phosphorylation sites of the histone deacetylases HDAC4 (S632) [19] and HDAC2 (S394) [20,21], and an increase in the S166 phosphorylation of the histone lysine demethylase KDM1A [22]. This implies that the activation of these epigenetic modifiers precedes the acquisition of resistance. We therefore sought to investigate whether the inhibition of HDAC2, HDAC4, or KDM1A would restore the expression of PHLDA1 in lapatinib-resistant HER2^+^ breast cancer cells. 

### 2.3. HDAC Inhibition Synergises with Receptor Tyrosine Kinase Inhibitors 

4SC-202 (Domatinostat) is a selective class I HDAC inhibitor that displays inhibitory activity against HDAC1/2/3 and also KDM1A [23]. As PHLDA1 expression is repressed by HDAC activity in the acquisition of RTKi resistance, we reasoned that HDAC inhibition should restore PHLDA1 expression. Accordingly, the treatment of lapatinib-resistant breast cancer cell lines with 4SC-202 led to an increase in PHLDA1 levels (Figure 4A) and elevated H3K27ac at the *PHLDA1* locus (Appendix A). When combined with lapatinib, 4SC-202 produced a synergistic effect on cell proliferation (Figure 4B and Appendix A) and colony formation capacity (Figure 4C and Appendix A). 4SC-202 and lapatinib also had a synergistic effect on the proliferation of lapatinib-resistant SKBR3 cells (Appendix A). Lapatinib-resistant HCC1954 spheres cultured in lapatinib exhibited growth arrest when treated with 4SC-202, which was diminished when spheroids were transfected with PHLDA1 siRNA (Figure 4D,E). This suggests that combining 4SC-202 and lapatinib may limit the growth of lapatinib-resistant HER2^+^ tumours by re-sensitising cancer cells to lapatinib through the re-expression of PHLDA1 (Figure 5). 

## 3. Discussion

The clinical efficacy of drugs targeting oncogenic RTKs is often limited by the emergence of resistance [24,25]. We previously identified PHLDA1 downregulation as a crucial step in the acquisition and maintenance of resistance to RTK-targeted therapy and demonstrated that the recovery of PHLDA1 expression could restore sensitivity to RTK inhibitors [7]. Here, we present a novel strategy to circumvent lapatinib resistance in HER2^+^ breast cancer by using a clinically relevant HDAC inhibitor to restore PHLDA1 expression. 

We have shown that PHLDA1 expression is downregulated in lapatinib-resistant HER2^+^ breast cancer cells and can be depleted in parental cell populations by treatment with lapatinib for 24 h. The treatment of lapatinib-resistant SKBR3 and HCC1954 cells with the phorbol ester PMA led to the recovery of PHLDA1 expression, which was abolished by co-treatment with inhibitors of PKC and ERK1/2. The regulation of endogenous PHLDA1 expression in HER2^+^ breast cancer cells is therefore likely to be controlled largely by PKC/MAPK signalling, consistent with observations in immortalised human mammary epithelial cells [13] and MEK1 mutant cancers [15]. 

We have previously shown, through microarray analysis, that the downregulation of PHLDA1 in RTKi-resistant cells is mediated by a suppression of PHLDA1 transcription [7]. To interrogate the molecular mechanism underlying this transcriptional repression, ChIP-qPCR analysis was used to identify differences in histone marks at the *PHLDA1* locus in lapatinib-resistant cells relative to matched parental lines. This revealed a significant decrease in the activatory marks H3K4me3 and H3K27ac at the *PHLDA1* locus in resistant cells, indicating that PHLDA1 downregulation is mediated at least in part by epigenetic changes. We suggest that this results primarily from the dampening of MAPK signalling downstream of RTK inhibition, since the treatment of parental HCC1954 cells with the MEK1/2 inhibitor PD0325901 led to a decrease in H3K4me3 and H2K27ac comparable to that induced by lapatinib. The treatment of FGFR2-driven MFE-296 endometrial cancer cells with the FGFR inhibitor PD173074 was also shown to decrease H3K4me3 and H3K27ac at the *PHLDA1* locus, and levels of PHLDA1 in this model were also found to be regulated by a PKC/MAPK-mediated pathway. The mechanism of PHLDA1 transcriptional repression described in this study may therefore be a common resistance mechanism to targeted therapies for RTK-driven cancers. 

Phospho-proteomic analysis revealed that HDAC2, HDAC4, and KDM1A were differentially phosphorylated in PD173074-treated MFE-296 cells compared to vehicle-treated controls. Since HDAC2 and HDAC4 catalyse the deacetylation of lysine residues on histones H2A, H2B, H3, and H4, and KDM1A catalyses demethylation at H3K4me and H3K9me, we hypothesised that the differential phosphorylation of these epigenetic modifiers, following RTK inhibition, may account for the decrease in H3K4me3 and H3K27ac at the *PHLDA1* locus observed in our breast and endometrial cancer models. In line with this, the treatment of lapatinib-resistant SKBR3 and HCC1954 cells with the clinically relevant class I HDAC inhibitor 4SC-202 [26] led to an increase in PHLDA1 levels. 4SC-202 inhibits HDAC1/2/3 and KDM1A [23], and ChIP-qPCR analysis showed that treatment with 4SC-202 led to an increase in H3K27ac at the *PHLDA1* locus. Whilst 4SC-202 did not affect H3K4me3 levels at the *PHLDA1* locus, it is possible that the simultaneous inhibition of KDM1A and HDAC1/2/3 by 4SC-202 may indirectly contribute to the increase in the PHLDA1 expression observed. 

The treatment of lapatinib-resistant SKBR3 and HCC1954 cells with PMA resulted in a far greater increase in PHLDA1 expression than that induced by 4SC-202. The regulation of PHLDA1 transcription is unlikely to be controlled solely by epigenetic changes and the modulation of transcription factor function downstream of ERK1/2 signalling may account for the greater increase in PHLDA1 expression observed following treatment with PMA. 

The recovery of PHLDA1 levels in RTKi-resistant cell lines restores sensitivity to RTK-targeted therapy [7]. We therefore hypothesised that the treatment of lapatinib-resistant breast cancer cell lines with 4SC-202 might restore sensitivity to HER2-targeted therapy. Bliss independence analysis showed 4SC-202 and lapatinib to exert synergistic effects on the growth of both parental and lapatinib-resistant cell lines. Equally, 4SC-202 was able to re-sensitise lapatinib-resistant spheres, dependent on PHLDA1 expression. The precise mechanism through which PHLDA1 induction restores sensitivity to lapatinib in HER2^+^ breast cancer models requires further investigation. The synergism between HDAC inhibitors and lapatinib in breast cancer has been demonstrated previously, through a mechanism involving Bim1 expression [27]. However, in the context of PHLDA1, we and others have shown PHLDA1 to negatively regulate AKT [7,8,9], and a resurgence of PI3K/AKT signalling has been reported to underlie resistance to HER2-targeted therapy [5]. An increase in cellular PHLDA1 levels may therefore dampen mitogenic signalling pathways downstream of AKT, leading to an inhibition of cell growth and survival. The data also suggest that PHLDA1 directly inhibits the activation of HER2 by disrupting its interaction with HER3 [8]. It is therefore probable that PHLDA1 suppresses survival signalling in lapatinib-resistant cells through the inhibition of HER2 signalling at multiple levels. 

In summary, we demonstrate that tumour cells resistant to targeted RTK therapy can be re-sensitised with the application of HDAC inhibitors. This is mediated through the reversal of PHLDA1 suppression, which underpins resistance in these models. Combination strategies involving RTK and HDAC inhibitors may therefore improve the longevity of targeted therapy by circumventing resistance. 

## 4. Methods and Materials

### 4.1. Cell Culture

Cells were obtained from the following suppliers: MFE-296 cells (Health Protection Agency, HPA, Porton Down, UK); AN3CA, SKBR3, and HCC1954 (American Type Culture Collection, ATCC, Teddington, UK) (Appendix A). All cell lines were cultured according to the recommended guidelines of the supplier. SKBR3^LapR^, HCC1954^LapR^, AN3CA^AZDR^, and MFE-296^AZDR^ cells were generated as described previously [7]. 

### 4.2. Spheroid Assay

HCC1954 spheres were formed from 1000 cells in 2.5% (*v*/*v*) methylcellulose (M0512, Sigma, Gillingham, UK) hanging droplets overnight at 37 °C. The spheres were then collected by centrifugation at 400 rcf for 4 min and then suspended in a collagen–Matrigel mix (2 mg/mL Collagen I (35429, Corning, Amsterdam, Netherlands), 17.5% Matrigel (354234, Corning), 15 mM HEPES, prepared in cell culture medium and buffered to physiological pH with 1 N NaOH). The sphere-containing gel mix was then seeded into a 96-well plate, to give approximately six spheres per well, and allowed to set at 37 °C. The spheres were cultured for 72 h before images were acquired using a Zeiss Axiovert 135. The area of each individual sphere was calculated using ImageJ software (Version 1.53, NIH, Bethesda, MD, USA). 

### 4.3. Western Blotting

Cell lysates were prepared using RIPA buffer (20–188, Millipore, Watford, UK) supplemented with protease and phosphatase inhibitor cocktails (524627, Millipore). Denatured proteins (20–40 μg) were separated by electrophoresis on 10–18% Tris–glycine gels. The proteins were subsequently transferred onto nitrocellulose membranes, blocked with 5% milk, and incubated with primary antibody diluted according to the manufacturer’s guidelines in 5% BSA/PBS. The membranes were then incubated with a species-appropriate horseradish peroxidase (HRP)-conjugated secondary antibody (DAKO, Cambridge, UK) before bands were visualised using enhanced chemiluminescence detection (GE Healthcare, Amersham, UK). 

### 4.4. Antibodies and Reagents

Rabbit anti-PHLDA1 (ab133654) was from Abcam (Cambridge, UK). Mouse anti-HSC70 (sc-7298) was from Santa Cruz (Heidelberg, Germany). Mouse anti-AKT (9220S), mouse anti-ERK1/2 (4696S), rabbit anti-p-AKT (Ser473) (9271S), rabbit anti-p-ERK (Thr202/Tyr204) (9101S), rabbit anti-acetyl-histone H3 (Lys27) (8173P), rabbit anti-histone H3 (9715), rabbit p-STAT3 (Tyr705) (9145S), and mouse anti-STAT3 (9139) were from Cell Signalling Technology (Leiden, Netherlands). Lapatinib, ZSTK474, GF109203X, TAK632, ruxolitinib, FR180204, PMA, and 4SC-202 were purchased from Selleckchem (Houston, TX, USA). U73122 and AKT-VIII were purchased from Sigma. 

### 4.5. Luciferase Assay

Firefly luciferase vector pGL4.19 harbouring 2.2 kb of the *PHLDA1* promoter was constructed as described previously [28]. Renilla luciferase control construct pRL-SV40P was obtained from Addgene (#27163). HCC1954 cells were seeded into a 96-well plate at a density of 10,000 cells per well 16 h prior to transfection with the firefly luciferase reporter construct and/or Renilla luciferase control construct. Transfection was performed using Lipofectamine 3000 reagent (L3000, Fisher, Paisley, UK). Then, 24 h after transfection, the cells were treated with DMSO or 40 µM FR180204 for a further 24 h. The cells were then lysed and the activity of the firefly and Renilla luciferase were measured using a Dual-Luciferase^®^ Reporter Assay System (E1910, Promega, Cambridge, UK) according to the manufacturer’s guidelines. Within each well, the activity of the firefly luciferase was normalised to that of the Renilla luciferase control. 

### 4.6. Two-Dimensional Proliferation Assay

Cells were seeded into 96-well plates at a density of 1500 cells/well. After 16 h, the cells were treated with 4SC-202 and/or lapatinib as indicated and maintained at 37 °C, 5% CO_2_. The treatment was refreshed after 3 days. After 5 days, the plates were transferred to an IncuCyte^®^ ZOOM imaging system and each well was imaged at four positions using a Nikon 10x objective. Three technical replicates were performed for each condition and each experiment was performed on three separate occasions. Cell confluence was analysed using IncuCyte ZOOM software (version 20151.2.5599). For each condition, the mean confluence was normalised to that of wells treated with DMSO. 

Synergistic drug combinations were identified using the Bliss Independence Model [29]. The expected additive effect of each drug combination (Bliss score) was determined using Equation (1). The Bliss score was then subtracted from the observed fractional inhibition at each drug combination to yield Excess Above Bliss scores, where >0 represents synergism. Equation (1):Bliss score = (Fractional inhibition compound A) + (Fractional inhibition compound B) − (Fractional inhibition compound A × Fractional inhibition compound B).(1)

### 4.7. Colony Formation Assay

Cells were seeded in six-well plates at a density of 1000 cells/well and treated after 24 h with varying combinations of lapatinib and 4SC-202. The cells were maintained for 21 days at 37 °C, 5% CO_2_, with drug treatment refreshed every 2–3 days. The colonies were fixed and stained using 0.1% crystal violet in methanol. The wells were photographed and colonies highlighted in pink using ImageJ. 

### 4.8. Lentiviral Vectors and Infection 

HCC1954 cells constitutively expressing HA-ERK2(GOF) were generated by transduction with lentiviral particles containing pBABEpuro-HA-ERK2(GOF) (Addgene #53203) [30]. The transduced cells were selected through culture in 1 μg/mL puromycin for 14 days prior to use.

### 4.9. siRNA Transfection

Cells were transfected with siRNA using lipofectamine 3000 (Fisher) following the manufacturer’s guidelines. SMART Pool siRNA containing five siRNA duplexes targeted against PHLDA1 (M-012389-01-0005) was purchased from Horizon Bioscience (Cambridge, UK).

### 4.10. Mass Spectrometry

MFE-296 cells were plated at in 10 cm dishes and either left untreated or cultured with either DMSO or 1 μM PD173074. The cells were lysed at 1, 7, or 14 days of culture. Cell lysis, digestion, solid-phase extraction, TiO_2_ metal oxide affinity chromatography, nanoflow liquid chromatography–tandem mass spectrometry, and identification and quantification of phospho-peptides were performed as previously described [7]. Dataset identifier: PXD008859. 

### 4.11. RNA Extraction and qPCR Analysis

RNA from the cultured cells was collected using the Monarch Total RNA Miniprep kit (T2010, New England Biolabs, Cambridge, UK), following the manufacturer’s guidelines. In total, 500 ng of RNA was converted to cDNA using LunaScript RT SuperMix (E3010, New England Biolabs) according to the manufacturer’s guidelines. qPCR analysis of cDNA was then performed using Luna Universal qPCR Master Mix (M3003, New England Biolabs) on a StepOnePlus Real Time PCR system (ThermoFisher Scientific, Paisley, UK), following the manufacturer’s guidelines. The qPCR primers used were as follows: PHLDA1 FW: ACCAAATACCGCACCCAC, RV: AGAAATGTGCTCGTCCCAC; Actin FW: AGAGCTACGAGCTGCCTGAC, RV: AGCACTGTGTTGGCGTACAG. 

### 4.12. ChIP-qPCR

The cells were fixed for 5 min in culture medium containing 1% formaldehyde, and then lysed in SDS lysis buffer (50 mM Tris pH 8.0, 1% SDS, 10 mM EDTA, protease inhibitor cocktail (Millipore)). The lysate was sonicated (Diagenode Bioruptor, Denville, NJ, USA) for 20 cycles (30 s on/30 s off) and 40 µg chromatin was incubated with either 2 µg anti-H3K4me3 (Diagenode, C15410003-50) or 5 µg anti-H3K27ac (Active Motif, 39133, Carlsbad, CA, USA) antibody at 4 °C for 16 h. Immunoprecipitation was subsequently performed using Dynabeads Protein G (Thermo Fisher). Immunoprecipitated chromatin was treated with RNAse A and proteinase K for 16 h prior to isolation by phenol–chloroform extraction and ethanol precipitation. The samples were then analysed by qPCR alongside a standard curve of known quantities of input DNA, with the enrichment of PCR reactions calculated relative to the negative control primers (Appendix A). 

### 4.13. Statistical Analysis

All data were analysed in GraphPad Prism software (version 9), with relevant statistical tests indicated in the figure legends. All data are presented as mean ± SEM. Generally, *t*-tests were conducted for tests between two samples, while ANOVA with multiple comparison tests were performed on tests with three or more samples. 

## Figures and Tables

**Figure 1 ijms-24-06228-f001:**
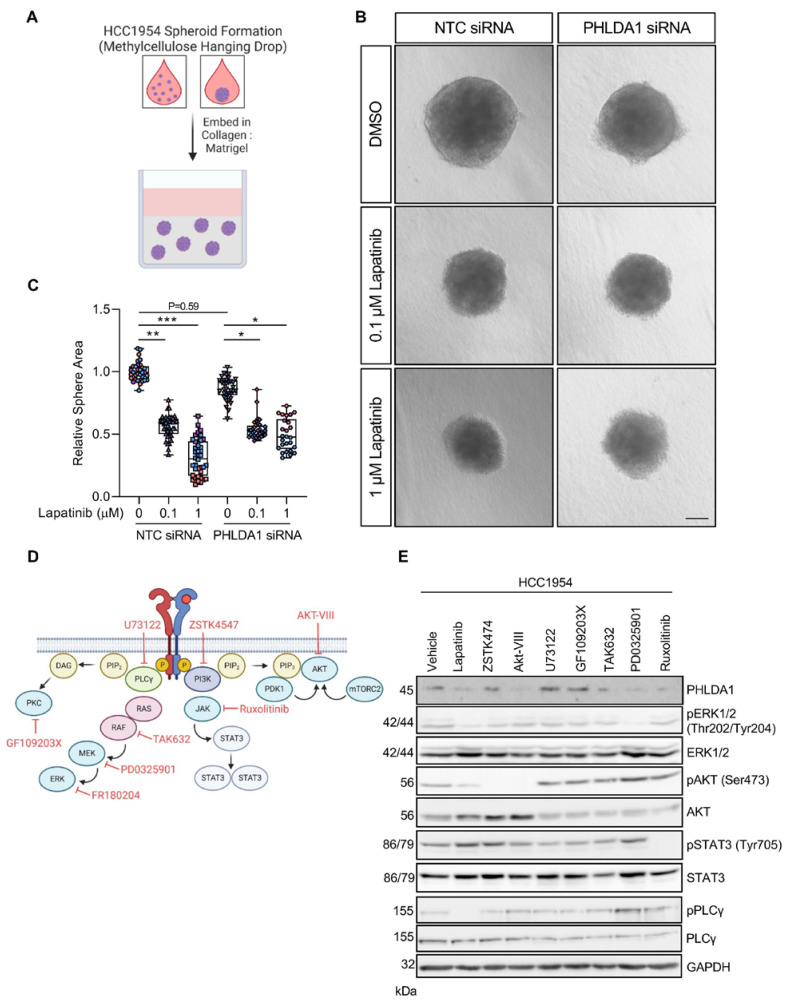
PHLDA1 is downregulated following inhibition of MEK1/2, AKT, RAF, and JAK. (**A**) Schema of spheroid growth model. (**B**) Brightfield images of HCC1954 spheres embedded in collagen–Matrigel and cultured for 72 h in lapatinib (0.1 or 1 µM) following transfection with either a non-targeting control (NTC) or PHLDA1 siRNA. (**C**) Quantification of sphere presented in (**B**). Individual colours on graphs indicative of technical replicates within each biological replicate. Scale bar = 100 µm. * *p* < 0.05, ** *p* < 0.01, *** *p* < 0.001. One way ANOVA with multiple comparison test. (**D**) Illustration of HER2 downstream signalling pathways. PI3K: phosphoinositide 3-kinase, PIP_2_: phosphatidylinositol (4, 5) bisphosphate, PIP_3_: phosphatidylinositol (3, 4, 5)-trisphosphate, PDK1: phosphoinositide-dependent kinase-1, mTORC2: mTOR Complex 2, PLCγ: phospholipase C gamma, DAG: diacylglycerol, PKC: protein kinase C, PKD: protein kinase D, JAK: Janus kinase. (**E**) Western blot analysis of PHLDA1 expression in HCC1954 cells treated for 24 h with inhibitors of key signalling pathways downstream of HER2: lapatinib (1 µM): HER2/EGFR inhibitor, ZSTK474 (1 µM): class I PI3K inhibitor, AKT-VIII (1 µM): AKT1/2 inhibitor, U73122 (10 µM): PLC inhibitor, GF109203X (5 µM): PKC inhibitor, TAK632 (5 µM): pan-RAF inhibitor, PD0325901 (200 nM): MEK1/2 inhibitor, ruxolitinib (3 µM): JAK1/2 inhibitor. Bands are representative of three independent experiments.

**Figure 2 ijms-24-06228-f002:**
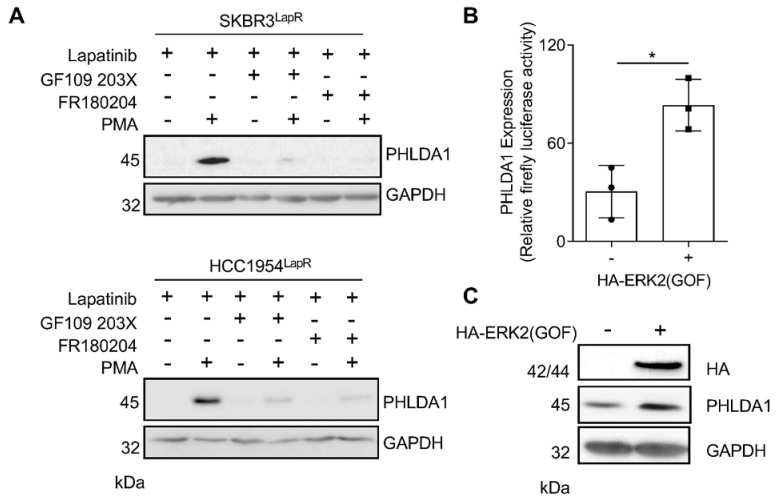
PHLDA1 transcription is regulated by PKC and ERK1/2. (**A**) Western blot analysis of PHLDA1 expression in lapatinib-resistant SKBR3 and HCC1954 cells (SKBR3^LapR^ and HCC154^LapR^) cultured in 1 µM lapatinib and treated for 48 h with 20 nM PMA in the presence and absence of 5 µM GF109203X and 40 µM FR180204. (**B**,**C**) PHLDA1 expression measured by luciferase activity (**B**) and Western blot (**C**) in parental HCC1954 cells and HCC1954 cells expressing HA-ERK2 (GOF) transfected with a PHLDA1 luciferase reporter construct. (**C**) Data are presented as mean ± SEM of three independent experiments. * *p* < 0.05, unpaired two-tailed *t*-test.

**Figure 3 ijms-24-06228-f003:**
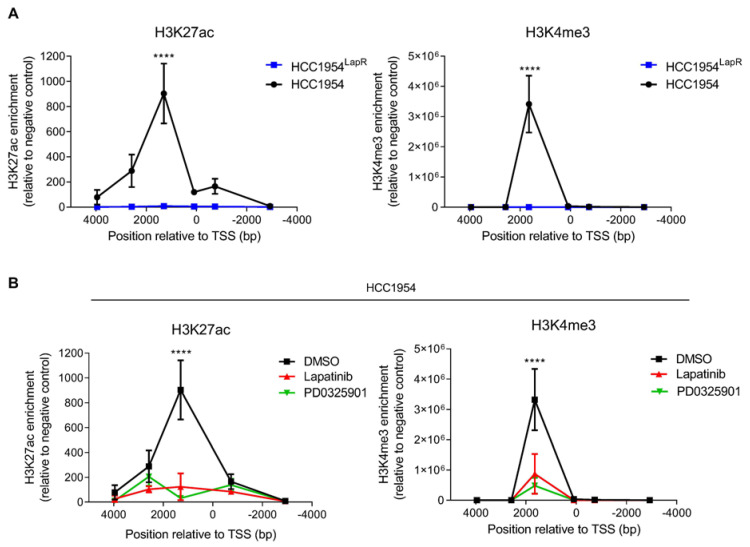
PHLDA1 downregulation is associated with epigenetic changes at the *PHLDA1* locus. (**A**) ChIP-PCR analysis of H3K27ac (**left**) and H3K4me3 (**right**) deposition at the *PHLDA1* locus in HCC1954 and HCC1954^LapR^ cells. (**B**) HCC1954 cells were treated with 1 µM lapatinib or 200 nM PD0325901 or DMSO for 72 h before H3K27ac (**left**) and H3K4me3 (**right**) at the *PHLDA1* locus was analysed by ChIP-PCR. Data are presented as mean ± SEM of three independent experiments. **** *p* < 0.0001. Two-way ANOVA with multiple comparison test.

**Figure 4 ijms-24-06228-f004:**
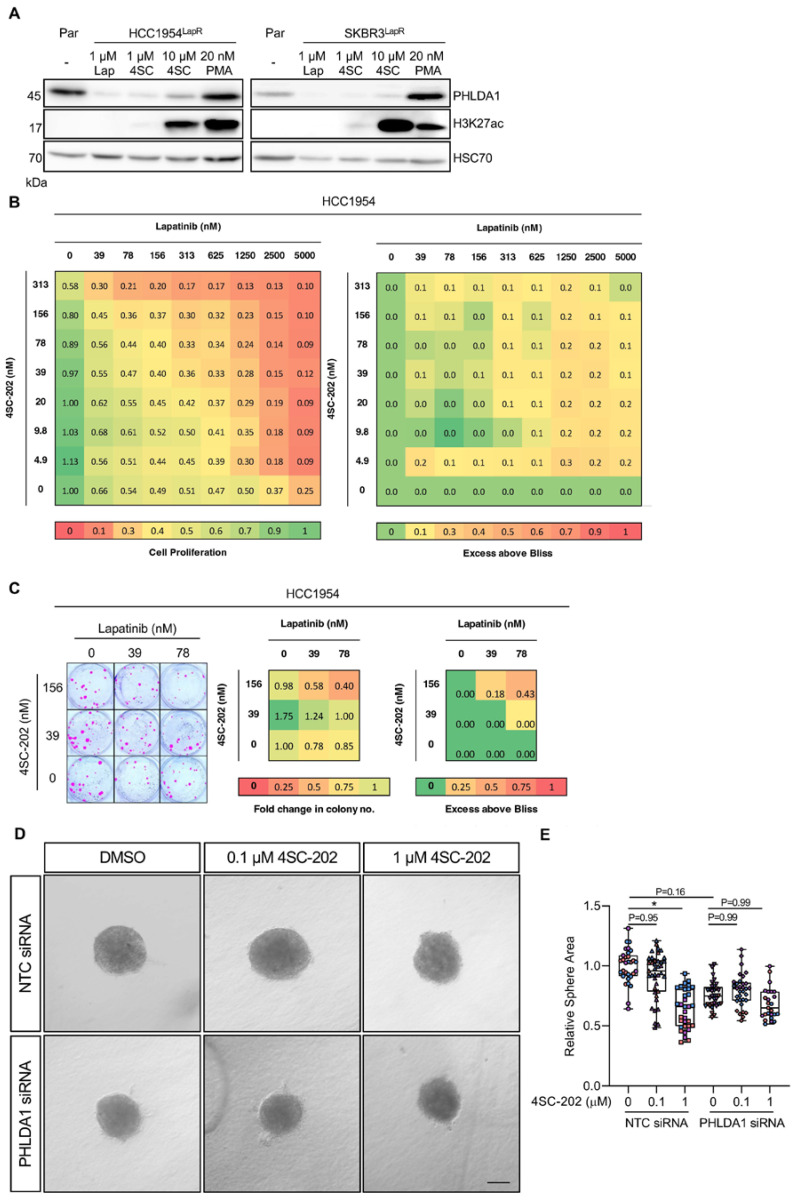
HDAC inhibitor 4SC-202 restores PHLDA1 expression in lapatinib-resistant cells and acts synergistically with lapatinib to inhibit cell proliferation. (**A**) Western blot analysis of PHLDA1 expression and H3K27ac in parental and lapatinib-resistant populations of HCC1954 and SKBR3 cells. Resistant cells grown in 1 μM lapatinib were treated with 4SC-202 or PMA for 48 h. (**B**) HCC1954 cells treated with varying combinations of 4SC-202 and lapatinib for 5 days before cell confluence was measured using an IncuCyte platform. Left panels, cell confluency fraction relative to vehicle control. Right panels, Bliss independence analysis of cell proliferation data, where Bliss score >0 represents synergy. (**C**) HCC1954 cells were seeded at low density into six-well plates and cultured for 21 days in varying combinations of 4SC-202 and lapatinib. Colonies (shown in pink) were visualised by crystal violet (left panel). Middle panel, normalised colony formation data showing mean fold change. Right panel, Bliss independence analysis. Data shown are the mean of three independent experiments. (**D**) Brightfield images of lapatinib-resistant HCC1954 spheres embedded in collagen–Matrigel and cultured for 72 h in 1 µM lapatinib with and without 4SC-202 (0.1 or 1 µM) following transfection with either a non-targeting control (NTC) or PHLDA1 siRNA. (**E**) Quantification of sphere images presented in (**D**). Individual colours on graphs indicative of technical replicates within each biological replicate. Scale bar = 100 µm. * *p* < 0.05. One way ANOVA with multiple comparison test.

**Figure 5 ijms-24-06228-f005:**
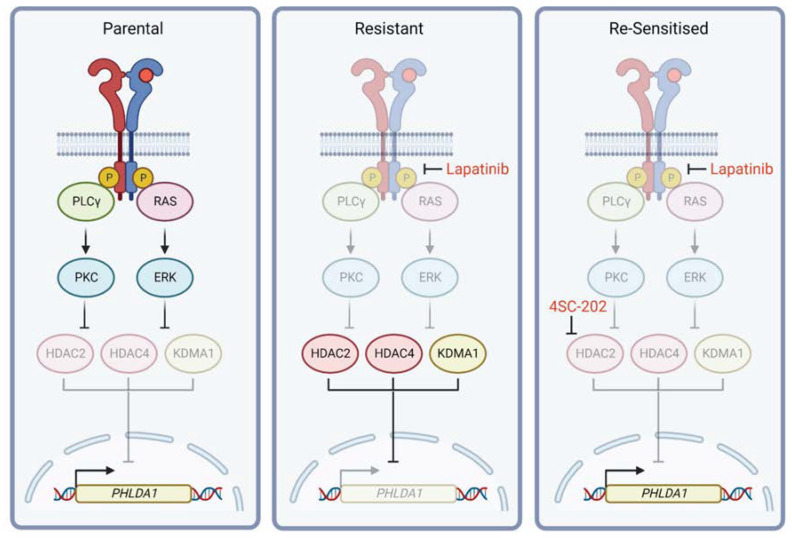
Proposed mechanism of PHLDA1 transcriptional regulation downstream of RTK activation. Stimulation of PLCγ and the MAPK pathway following receptor activation leads to increased activity of PKC and ERK, which in turn promotes phosphorylation and inhibition of HDAC2 (pS394), HDAC4 (pS633), and KDM1A (pS166). In resistant cells, these HDACs become active, repressing PHLDA1 transcription and allowing continued cell proliferation. Combined treatment with an HDAC inhibitor (4SC-202) recovers PHLDA1 transcription, re-sensitising cells to targeted therapy.

## Data Availability

Mass spectrometry data used in this study is published on Proteome Xchange with the identifier PXD008859. All other data available upon request.

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
