# Peer review of "HDAC Inhibition Restores Response to HER2-Targeted Therapy in Breast Cancer via PHLDA1 Induction"

_ijms, 2023, doi:10.3390/ijms24076228_

Round 1

Reviewer 1 Report

Although the manuscript is well written and represented following points need to be addressed to meet the publication standards.

Major comments

1.     More literature should be cited or incorporated to define a clear background understanding and provide rationality.

2.     The use of AN3CA and Ishikawa cells in the manuscript are not breast cancer cell lines but endometrial cancer cell lines. How does the work specifically relate to breast cancer under this condition? Also, why the use of these multiple cell lines has been done?

3.     Including specific clinical parameters is needed to support the notion of cell lines and drug selection. The clinicopathological and molecular profiles of the cell lines should be incorporated in the manuscript as supplementary information.

4.     The use of PHLDA1 as a target and the significance of selecting this particular gene expression induction must be discussed in more detail.

5.     Detailed information on reagents or antibodies must be incorporated in the manuscript, such as catalog numbers, company names, etc.

6.     The materials and methods must be clearly described and superficially mentioned.

7.     The HDAC inhibitor needs to be discussed more clearly,

8.     The rationale behind selecting the drug combination should also be explained with some details to understand the mechanism and effects of the drug in combination.

9.     In lines 22-24, the author states, “We have identified inhibition of MAPK signalling as a crucial step in PHLDA1 down-regulation and show that MEK1/2 inhibition produces significant epigenetic changes at the PHLDA1 locus, specifically, a decrease in the activatory marks H3Kme3 and H3K27ac” how did they infer this?

10.  There should be more details regarding the statistical methods used in the study. Only line 278 needs to be sufficient for the readers to understand and interpret the protocol.

11.  The conclusions should be well established and very rushed statement as in line 142 where the authors state that “we suggest that this results primarily from the dampening of MAPK signalling 142 downstream of RTK inhibition, since treatment of parental HCC1954 cells with the MEK1/2 inhibitor PD0325901 led to 143 a decrease in H3K4me3 and H2K27ac comparable to that induced by lapatinib.” More details need to be incorporated as supplementary information.

12.  Also, “The mechanism of PHLDA1 transcriptional repression described in this study may therefore be common to 147 multiple different cancer types” which other cancers and how?

13.  More in vivo or ex vivo data may help make a generalized conclusion (title).

14.  Likewise, many assumptions and conclusions made in the discussion are based on hypotheses and must be evaluated with considerable data and more referencing.

Minor comments

    1.    There needs to be a protein ladder (marker) in the western blot and figures.

    2.    The densitometry analysis of the blots needs to be incorporated into the manuscript.

    3.    More scientific and grammatical language should be implemented.

    4. Many phrases are common in certain other manuscripts. Therefore, it is suggested that the authors check the manuscript on online portals such as iThenticate.

Reviewer 3 Report

Dear author

Thank you of the submission of your article to our journal. HER2-positive breast cancer once annoyed many clinicians due to its hyperproliferative nature. The advent of trastuzumab dramatically changed the treatment of HER2-positive breast cancer. After that, lapatinib, pertuzumab, T-DM1, and T-DXd appeared in the clinical setting, and the overall survival of HER2-positive breast cancer were significantly improved regardless of whether they were primary or metastatic breast cancer. On the other hand, the emergence of resistance to anti-HER2 drug therapy has also become a major clinical problem, and many basic studies have been conducted so far. This study also focuses on the fact that PHLDA1 is greatly involved in the treatment resistance of HER2-positive breast cancer, and although it is basic research, it is very useful in that it clearly shows that resistance can be overcome by HDAC inhibitors. This study will be of great interest to the readers of our journal. Please correct only one point described below.

Fig.1

“Schematic of spheroid growth model” should be revised to “Schema of spheroid growth model”a
